# *SOD2*, a Potential Transcriptional Target Underpinning CD44-Promoted Breast Cancer Progression

**DOI:** 10.3390/molecules27030811

**Published:** 2022-01-26

**Authors:** Nouralhuda Alateyah, Ishita Gupta, Radoslaw Stefan Rusyniak, Allal Ouhtit

**Affiliations:** 1Biological Sciences Program, Department of Biological and Environmental Sciences, College of Arts and Sciences, Qatar University, Doha P.O. Box 2713, Qatar; na1601400@student.qu.edu.qa (N.A.); rusyniak@qu.edu.qa (R.S.R.); 2Department of Basic Medical Science, College of Medicine, QU Health, Qatar University, Doha P.O. Box 2713, Qatar; ishita.gupta@qu.edu.qa; 3Biomedical and Pharmaceutical Research Unit, QU Health, Qatar University, Doha P.O. Box 2713, Qatar

**Keywords:** *SOD2*, CD44, hyaluronan, breast cancer, invasion

## Abstract

CD44, a cell-adhesion molecule has a dual role in tumor growth and progression; it acts as a tumor suppressor as well as a tumor promoter. In our previous work, we developed a tetracycline-off regulated expression of CD44’s gene in the breast cancer (BC) cell line MCF-7 (B5 clone). Using cDNA oligo gene expression microarray, we identified *SOD2* (superoxide dismutase 2) as a potential CD44-downstream transcriptional target involved in BC metastasis. *SOD2* gene belongs to the family of iron/manganese superoxide dismutase family and encodes a mitochondrial protein. *SOD2* plays a role in cell proliferation and cell invasion via activation of different signaling pathways regulating angiogenic abilities of breast tumor cells. This review will focus on the findings supporting the underlying mechanisms associated with the oncogenic potential of *SOD2* in the onset and progression of cancer, especially in BC and the potential clinical relevance of its various inhibitors.

## 1. Introduction

Tumor cell invasion is the most defining and recurring event during the multistage process of metastasis. It involves the activation of a complex molecular network involving, involving, in particular, cell adhesion molecules (CAMs) [1,2] which facilitate the adhesion of invading cells to their surrounding extracellular matrix (ECM) [3]. Among numerous CAM protein families, CD44 is the principal cell surface receptor for hyaluronic acid (HA), a major component of the ECM produced by embryonic stem cells, connective tissue cells, bone marrow cells [4,5], and cancer cells [6,7]. We have previously reported that the interaction of CD44 with HA activates various oncogenic signaling pathways, leading to tumor cell survival, proliferation, and invasion [8,9,10,11,12,13].

In order to better understand the exact function of the standard form of CD44 (CD44s) in BC and further elucidate the signaling mechanisms that underpin CD44-promoted BC invasion/metastasis, we have previously developed a tetracycline (Tet)-Off-regulated expression system of CD44s both in vitro [11] and in vivo [12]. Moreover, we applied microarray gene expression profiling, using total RNA samples isolated from “CD44-On versus CD44-Off cells”. Bioinformatics analyses revealed more than 200 CD44-downstream transcriptional target genes that were upregulated or downregulated. Four genes have already been validated, and their signaling pathways linking their transcriptional activation to CD44 upstream regulation, have already been published [14].

Among the remaining genes, *SOD2* was significantly upregulated upon interaction of CD44 with its major ligand hyaluronan (HA), suggesting that *SOD2* might be an additional novel transcriptional target of CD44-downstream signaling that underpins its role in promoting BC tumor cell invasion and metastasis.

*SOD2* encodes the manganese superoxide dismutase (Mn-SOD), which is involved in regulating oxidative stress in the cell through the dismutation of superoxide radicals (O_2_−) to H_2_O_2_ and molecular oxygen [15,16,17]. Reactive oxygen species (ROS) cause DNA damage, resulting in genetic aberrations and genomic instability, thus promoting mutagenesis and carcinogenesis [18,19]. Damage by ROS is usually mitigated by antioxidant actions of non-enzymatic antioxidants or antioxidant enzymes, thereby reducing the likelihood of mutations and possibly oncogenic transformation [20]. Alteration in the expression of antioxidant enzymes, like catalase (CAT), glutathione peroxidases (GPx), and superoxide dismutases (SOD), can impair regulation of enzymatic activity and modify ROS detoxification [21].

The SOD family encompasses three members, SOD1, *SOD2*, and SOD3. While SOD1 is localized in the cytoplasm and the mitochondrial intermembrane space, *SOD2* is exclusively present in the mitochondrial matrix, and SOD3 is extracellularly present [22]. Genetic variations in *SOD2* have been found to be associated with many diseases, including neurodegeneration, mitochondrial dysfunction, premature aging, angiogenesis, and cancer [23,24,25,26,27].

This review presents data from the literature supporting our hypothesis that *SOD2* might be a novel transcriptional target of CD44-downstream signaling, which promotes BC cell invasion. We have discussed these lines of evidence and proposed a model of the signaling pathways linking CD44 activation by HA to the transactivation of *SOD2* to promote breast tumor cell invasion.

## 2. Structure of *SOD2*

The human *SOD2* gene, located on the long arm of chromosome 6 (6q25.3; from base pair 159,669,069 to base-pair 159,762,281), consists of 5 exons and 4 introns [16,17] encoding 222 amino acids [22,28,29]. The genetic sequence reveals that the *SOD2* promoter lacks the typical TATA and CAAT boxes; it is, however, rich in GC sequences as well as binding sites for specificity protein (SP) and activator protein 2 (AP2) transcription factors [29]. Sp1 and NF-kB are two transcription factors that have been well-studied in the regulation of *SOD2* expression. Sp1 maintains *SOD2’*s basal promoter activity, whereas Nf-kB increases *SOD2* transcription mostly through a cis-element inside the second intronic enhancer region [30]. Additionally, the 3′-untranslated region of *SOD2* mRNA shows a conserved 41 bp translational enhancer region [30].

The *SOD2* protein is a homo-tetramer (86–88 kDa in total), with each of the four subunits having a molecular mass of around 23 kDa, and each including a manganese (Mn^+3^) cofactor [28]. The precursor monomer of *SOD2* starts with a 26 amino acid mitochondrial targeting sequence (MTS), which is vital for mitochondrial localization [31]. For example, *SOD2* gene polymorphisms, which result in a substitution of Ala-9 to Val (Val9Ala) [28], or Ala-16 to Val (Val16Ala) alter the structure of *SOD2* MTS, and have been linked with susceptibility to various pathologies [32,33].

Each mature *SOD2* monomer consists of two unique domains, an N-terminal helical hairpin domain and a C-terminal a/b domain. The C-terminal a/b domain contains five-alpha helices and three-stranded antiparallel beta-sheets. Together, these domains encompass key residues (D159, H163, H26, and H74), allowing the active site to interact with the Mn³^+^ cofactor and a water molecule [34], which are critical to the enzyme’s dismutase activity.

## 3. Functions of *SOD2*

Physiologically, *SOD2* is expressed in both normal and malignant cells, thus, the following sections will discuss the role of *SOD2* in both contexts.

### 3.1. Physiological Function of SOD2 in Normal Cells

While aerobic respiration produces low amounts of ROS, cells utilize SODs to prevent these molecules from causing damage. SODs provide such a mechanism, since most of the ROS is produced by the electron transport chain in the mitochondrial matrix of human cells, where *SOD2* is ideally localized [35]. Integration of the Mn cofactor into the *SOD2* catalytic site triggers the enzyme to carry out its dismutase activity [15,16], and thus enables it to maintain mitochondrial integrity [36,37] and protect mitochondrial DNA (mtDNA) and mtDNA Polymerase C from oxidative damage or inactivation [38].

### 3.2. Physiological Functions of SOD2 in Cancer

Various lines of evidence lead to the conclusion that *SOD2* can act either as a tumor suppressor gene (TSG) or as an oncogene [39].

Variations in *SOD2* activity are found during different cell cycle stages [40,41,42]; and an argument for a TSG role can be made based on studies showing reduced *SOD2* expression in pancreatic cancer cells, as well as glioma cells, and *SOD2* results showing significantly delayed tumor cell growth in nude mice xenograft studies [43,44]. Such loss of *SOD2* expression enhances cell cycle progression (mitosis) by increasing O_2_•^−^ levels [45], thus stimulating ROS-mediated DNA damage and inducing cell transformation and tumorigenesis [46,47].

Different mechanisms have been proposed for the loss of *SOD2* in cancer. Mutations in the *SOD2* gene promoter have been reported to impact the AP2 binding pattern in colorectal cancer cells, resulting in loss of *SOD2* expression [48]. Some evidence also points to epigenetic modifications as a reason for the downregulation of *SOD2* expression [39,41,42]. *SOD2* promoter methylation was reported in both transformed and cancer cells [49,50], and in another study, cytosine methylation of the second intron of *SOD2* was found to result in reduced *SOD2* expression [47]. Such methylation patterns have been reported in breast and pancreatic cancers as well as multiple myeloma [49,50,51,52].

While reduced *SOD2* expression is commonly reported in the early stages of tumor growth, some studies report elevated levels of *SOD2* during metastatic progression [36,53,54]. *SOD2* was overexpressed in metastatic tumor lesions and aggressive tumor cell lines. Upregulated *SOD2* levels enhance the oxidation of PTEN and PTP-N12, further activating Akt and p130 cas/Rac1 signaling pathways, respectively [37]; these pathways regulate the hallmarks of cancer during tumor progression [37,55,56,57,58]. Thus, these data put together point to an oncogenic role of *SOD2* in the onset and progression of cancer.

Table 1 summarizes the mechanisms underlying the dual role of *SOD2* in different cancers.

### 3.3. Physiological Functions of SOD2 in Invasion, Metastasis, and Angiogenesis

Accumulation of free radicals is associated with mitochondrial function and metastatic disease progression. Although, ROS play a role in tumor initiation and progression, the underlying mechanisms and causes of enhanced ROS accumulation during tumorigenesis still lie nascent.

Findings from various studies strongly support our hypothesis that the activation of CD44 by its major ligand hyaluronan (HA) activates the transcription of *SOD2* to promote tumor cell invasion/metastasis (Figure 1).

*SOD2* expression increases cellular oxidant/antioxidant ratios, which are directly associated with tumor progression, angiogenesis and migration, and invasion [53,87]. In several cancers, *SOD2* expression is elevated during metastasis [61,62,88,89,90,91,92,93]. Increased *SOD2* expression in metastatic cancer cell lines as well as in tumor tissues is associated with increased *SOD2* activity, thus, indicating a functional role for *SOD2* during metastatic progression and poor outcome, particularly in BC [54,81,94,95,96], where elevated *SOD2* expression is associated with metastatic progression in estrogen receptor (ER) negative BCs [93].

The transcriptional repressor DDB2 plays an important role in the dual nature of *SOD2* in BC [65]. While in ER-positive BC cells (MCF7), DDB2 reduced acetylated H3 histones and decreased binding of Sp1 to *SOD2* promoter, in ER-negative BC cells (MDA-MB-231), the lack of DDB2 induced high levels of *SOD2* [65]. In the BC cell line, T47D, *SOD2* expression was regulated by progestin [97].

Additionally, oxygen-free radicals and ionizing radiation have been shown to stimulate *SOD2* expression in a TNFα-dependent manner [98,99], as altered TNFα levels correlate with enhanced metastatic disease [100]. Briefly, TNFα ligands bind to TNFRs to stimulate upstream IKKs and phosphorylate IκB that is bound to the inactive NF-κB dimer (RelA + p50) in the cytoplasm. IκB underwent proteasome degradation and activated NF-κB translocates into the nucleus, where it triggers the expression of genes encoding proteins. NFκB triggers the activity of the *SOD2* promoter, thus inducing *SOD2* expression [101,102]. NF-κB has also been found to cooperate with ZEB2 to increase *SOD2* gene expression in cells containing high levels of CD44 (CD44H), which leads to the induction of epithelial-mesenchymal transition mechanism [103].

Overexpression of *SOD2* in tumor cells correlates with enhanced expression and activity of MMPs, leading to enhanced matrix degradation and a release of cytokines and growth factors, thereby promoting metastasis [104]. For example, in the ER positive BC cell line, MCF7 increased ROS, enhanced *SOD2* expression, and activated MMP2 in parallel [105]. Similarly, overexpression of *SOD2* in HT-1080 fibrosarcoma, as well as 253J bladder carcinoma cells, significantly promoted migration and invasion both in vitro and in vivo [56,92,106]. One of the effects of increased *SOD2* expression is a significant reduction in catalase activity, resulting in increased H_2_O_2_ production [53,56], which has been shown to upregulate the expression of MMP-1 [107], and might increase the levels of VEGF and MMP-9, known contributors of cell invasion and angiogenesis [108,109].

Further, binding of Ets-1 to the MMP-1 promoter and sustained JNK signaling requires the enzyme activity of *SOD2* [108,110]. Thus, it appears that *SOD2* is essential for the stimulation of MMP-1 as well as other MMP family members via the H_2_O_2_-dependent activation of MAPK signaling [108,110,111], resulting in the remodeling and degradation of the ECM and the basement membranes, thereby promoting tumor cell invasion and metastasis [112].

The above, along with our findings that *SOD2* was significantly upregulated upon the interaction of CD44 with its major ligand HA [14]; prompted us to hypothesize that *SOD2* might be a novel transcriptional target of CD44-downstream signaling, and thus, have a critical role in promoting BC tumor cell invasion and metastasis.

As a matter of fact, elevated content of *SOD2* is correlated with increased tumor cell invasion, metastasis, proliferation, and resistance to apoptosis [53]. Thus, there is an association between CD44 and *SOD2* as they are involved in common signaling pathways promoting metastasis, cell migration, cell invasion, and angiogenesis [113]. Our initial experiments revealed that HA treatment of MDA-MB-231 BC cells significantly increased CD44 expression at 24 h, which was accompanied by a parallel pattern of expression of its transcriptional target *SOD2* at RNA levels. More interestingly, inhibition of CD44 using specific siRNA significantly decreased RNA expression levels of *SOD2*. Ongoing functional, molecular, and pharmacological experiments aim to validate the physiological relevance of the CD44/*SOD2* signaling pathway in BC cell invasion and further identify its intermediate molecular players. 

### 3.4. Potential Inhibitors Targeting SOD2

There is a dire need to develop anticancer therapeutics targeting ROS signaling, and hence, use of SOD mimetics that simulate endogenous SOD enzyme activity or inhibit oxidative injury—biomarkers of cell proliferation are being explored.

The epigenetic drug zebularine was found to effectively reverse *SOD2* promoter methylation in the KAS6/1 human multiple myeloma cells [49]. In addition, tirchostatin A and sodium butyrate were also found to alter *SOD2* expression by regulating histone methylation and acetylation in breast cancer cells [51,114]. Synthetic SOD mimetics were also reported to cross the blood–brain barrier and exert a protective effect against neurodegenerative diseases [115,116,117,118].

Adenoviral vectors, in combination with chemotherapeutic agents have also been used to induce SOD expression. In breast tumor xenograft models, injection of Cu-ZnSOD or *SOD2* adenoviral vectors in combination with BCNU significantly reduced breast cancer cell growth and increased nude mice survival [119]. Moreover, dual gene virotherapy utilizing adenovirus consisting of *SOD2* and TRAIL-induced apoptosis inhibited colorectal tumor growth in xenograft models [120].

As mentioned above, elevated ROS stress in cancer cells is an unfavorable event associated with increased cancer progression and carcinogenesis. Recently, research has identified drugs capable of increasing cellular ROS accumulation to eliminate cancer cells [121,122]. Since oxidative stress linked with carcinogenesis makes cancer cells highly dependable on their antioxidant systems [123,124,125], inhibition of the antioxidant system could result in a substitutional accumulation of ROS and provoke apoptosis. Thus pharmacological drugs could be used to eradicate cancer cells by applying significant ROS stress.

Promoting ROS accumulation in cancer cells to a toxic level is the focus of some emerging cancer therapies, which makes *SOD2* a good potential target. Various drugs are known to trigger ROS generation and accumulation; these include mitochondrial electron transport chain modulators (doxorubicin, topotecan), redox-cycling compounds (motexafin gadolinium), glutathione (GSH) depleting agents (buthionine sulphoximine, β-phenylethyl isothiocyanates (PEITC)), and inhibitors of SOD (2-methoxyestradiol) and catalase (e.g., 3-amino-1,2,4-triazole) [126,127,128].

Rapamycin and its analogs are being used in clinical trials for certain types of cancer as rapamycin alone can inhibit cell proliferation and induce apoptosis [129,130]. Moreover, rapamycin promotes chemosensitivity by blocking mTORC1 and activating Akt, thus resulting in ROS-induced cell death [131,132,133,134].

FOXM1 is a transcription factor that is over-expressed in the majority of human tumors. It can induce *SOD2* expression, resulting in a decrease of ROS-levels in cancer cells, thereby promoting angiogenesis and metastasis [135,136]. Targeting *FOXM1* in combination with oxidative stress is considered as an essential therapeutic strategy against cancer. Proteasome inhibitors (MG132) as well as thiazole antibiotics (siomycin-A) reduce *FOXM1* expression and activity and induce apoptosis through DNA damaging agents, including doxorubicin and γ-irradiation [137,138,139].

In contrast to the above, the use of antioxidants has also shown some positive results. These often take the form of traditional therapeutics—natural occurring antioxidants. Natural products including curcumin [140,141,142], lycopene [143], ginger [144,145], and hesperetin [146] are redox-active and induce a pro-oxidative mechanism of action indicating their role as anti-tumorigenic, anti-inflammatory, and anti-angiogenic.

In one study, curcumin oil was found to maintain *SOD2* expression and delayed bile acid-enhanced esophageal injury and cancer progression [147]. It has also been shown to decrease breast cancer-induced lung metastasis by approximately 40% and, in combination with paclitaxel, further inhibited breast cancer growth [148].

Astaxanthin, a xanthophyll, possesses anti-inflammatory properties and exerts its effect via ROS/reactive nitrogen species scavenging by inhibiting the expression of inflammatory cytokines (NF-κB, TNF-α, and IL-8) [149].

The natural dietary element, ginger has both antioxidant and anti-carcinogenic properties: ginger supplementation enhanced SOD, CAT, GPx, glutathione-S-transferase, and glutathione reductase activities, thus reducing oxidative stress as well as cancer cell growth [150].

Oroxylin A, a flavonoid extracted from a Chinese medicinal plant, can induce *SOD2* gene expression, glycolysis inhibition, and reduced tumor growth of transplanted human breast tumors in vivo [151].

One of the dietary combination supplements, protandim, was shown to be effective in inhibiting tumor progression as well lipid peroxidation [152]. Protandim enhanced activities of both SOD and CAT, thus reducing tumor growth [153].

This indicates the use of antioxidants to target ROS generation.

Table 2 summarizes anti-*SOD2*-based therapeutic approaches.

## 4. Conclusions

Several studies have suggested the complicity of antioxidant enzymes in cancer development and identified high expression of *SOD2* as a factor in cancer progression. The ability to use *SOD2* to modify the cellular redox environment can help with the identification of redox-responsive signaling events that stimulate malignancy, such as invasion, migration, and prolonged tumor cell survival. Additional studies of these redox-driven events will help in the development of targeted therapeutic strategies to efficiently restrict redox-signaling essential for malignant progression. In particular, results from our own work and others support our hypothesis that CD44-HA interaction can transactivate *SOD2*, ultimately leading to BC progression.

## Figures and Tables

**Figure 1 molecules-27-00811-f001:**
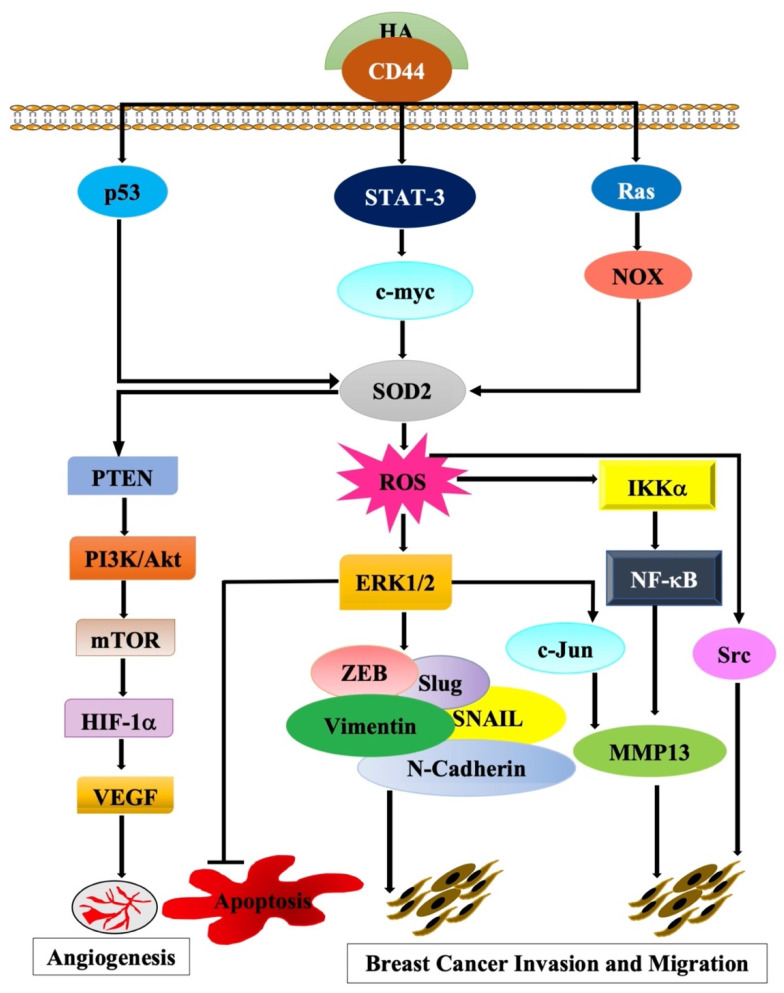
A proposed novel model of the molecular signaling pathways that link the activation of CD44 by its ligand hyaluronan (HA) to the transactivation of its potential target gene *SOD2*.

**Table 1 molecules-27-00811-t001:** Mechanisms underlying the expression of *SOD2* in different cancers.

Type of Cancer	*SOD2* Expression	Underlying Mechanism	References
Bladder	Downregulated	Not determined	[59]
Brain	Downregulated	*SOD2* hyperacetylation	[60]
	Upregulated	Not determined	[61,62,63]
Breast	Downregulated	p53 transcriptional inhibition Presence of *DDB2* Epigenetics *SOD2* hyperacetylation	[51,64,65,66]
	Upregulated	Loss of p53 NF-kB Nrf2 activation	[64,67,68]
Colorectal	Downregulated	Elevated AP1 at SNP promoter	[48]
	Upregulated	Not determined	[69,70]
Esophageal	Downregulated	Not determined	[71]
	Upregulated		[59]
Leukemia	Downregulated	Not determined	[59]
	Upregulated	ARNT activation	[44]
Liver	Downregulated	Calcium blocking of SIRT3	[72,73]
	Upregulated	Not determined	[59]
Lung	Downregulated	Ala16Val substitution	[74]
	Upregulated	Not determined	[59]
Lymphoma	Upregulated	Not determined	[75]
Melanoma	Downregulated	Loss of heterozygosity	[76]
Multiple Myeloma	Downregulated	Epigenetic silencing	[49,50]
Ovarian	Upregulated	Keap1 mutation Nrf2 activation	[52,57,77]
Pancreatic	Downregulated	Epigenetic silencing miR-301a activation Ala16Val substitution	[78,79,80]
	Upregulated	Not determined	[81]
Prostate	Upregulated	Low miR-17 expression	[70,82]
Renal Clear Cell	Downregulated	HIF-1α activation Oxidation	[83,84]
Sarcoma	Downregulated	Nitration	[59]
	Upregulated	Not determined	[59]
Tongue Squamous Cell	Downregulated	miR-222 activation	[85]
	Upregulated	c-myc activation	[86]

Ala: alanine; AP1: activated promoter 1; ARNT: aryl hydrocarbon receptor nuclear translocator; c-myc: cellular myelocytomatosis; DDB2: damaged DNA binding 2; HIF-1α: hypoxia-inducible factor 1-alpha; Keap1: kelch-like ECH-associated protein 1; miR: micro-RNA; NF-kB: nuclear factor-kappa B; Nrf2: nuclear factor erythroid 2–related factor 2; SIRT3: sirtuin 3; SNP: single nucleotide polymorphism; Val: Valine.

**Table 2 molecules-27-00811-t002:** Therapeutic Strategies for targeting SOD2.

Therapy	Treatment	Mode of Action	References
Epigenetic Drugs	Zebularine	Reverses *SOD2* promoter methylation	[49]
	Tirchostatin A	Regulates histone methylation and acetylation	[51,114]
	Sodium butyrate		
Adenoviral Vectors	Cu-Zn SOD/*SOD2* adenoviral vectors combined with BCNU	Induces SOD expression to reduce cancer cell growth and increases survival	[119]
	*SOD2* and TRAIL combination	Enhances SOD expression to stimulate cell apoptosis and reduce tumor growth	[120]
Topoisomerase inhibitors	Doxorubucin	Reduces cancer growth by binding to the topoisomerase enzymes and block topoisomerases 1 and 2	[154]
	Topotecan		
Redox Cyclin Compounds	Motexafin gadolinium	Induces apoptosis by alteration in mitochondrial membrane potential, depletion of intracellular GSH and increased ROS production.	[155]
GSH Depleting Agents	Buthionine sulphoximine	Eliminates GSH from the cells by blocking GPx and accumulates ROS production Oxidative inactivation of H-Ras and NF-kB Oxidative damage of mitochondria Induces apoptosis	[126,127,128,133]
	β-phenylethyl isothiocyanates (PEITC)		
*SOD2* inhibitors	2-methoxyestradiol	Blocks both manganese and copper, zinc superoxide dismutases and reduces cancer growth	[156]
Catalase	3-amino-1,2,4, triazole	Reduces CAT activity Enhances intracellular H_2_O_2_ levels Decreases GPx activity	[157]
Anti-fungal Antibiotic	Rapamycin	Inhibits cell proliferation Stimulates apoptosis Blocks mTORC1 and activates Akt to promote chemoresistance	[129,130,131,132,133,134]
Proteosome inhibitor	MG132	Reduces FOXM1 expression to induce apoptosis	[137,138,139]
Thiazole Antibiotic	siomycin-A		
Natural Products	Curcumin	Conserves *SOD2* expression Delays bile acid-induced esophageal injury and cancer progression	[147,148]
	Astaxanthin	Inhibits expression of NF-κB, TNF-α, and IL-8 to induce ROS/reactive nitrogen species scavenging	[149]
	Ginger	Inhibits oxidative stress and cancer growth by stimulating SOD, CAT, GPx, glutathione-S-transferase, and glutathione reductase activities	[150]
	Oroxylin A	Induces *SOD2* gene expression Inhibits glycolysis Suppresses tumor growth	[151]
	Protandim	Stimulates SOD and CAT activities Reduces tumor growth	[153]

2-ME: 2-methoxyestradiol; Akt: protein kinase B; BCNU: 1,3-bis(2-chloroenthyl)-1-nitrosurea; CAT: catalase; Cu-Zn: copper–zinc; FOXM1: forkhead box M1; GPx: glutathione peroxidases; GSH: glutathione; IL: interleukin; mTORC1: mammalian target of rapamycin complex 1; NF-kB: nuclear factor-kappa B; PEITC: phenylethyl isothiocyanates; ROS: reactive oxygen species; TNF: tumor necrosis factor; TRAIL: TNF-related apoptosis-inducing ligand.

## Data Availability

Data sharing is not applicable to this article as no new data or datasets were created, generated, or analyzed in this study.

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
