# Peer review of "SOD2, a Potential Transcriptional Target Underpinning CD44-Promoted Breast Cancer Progression"

_molecules, 2022, doi:10.3390/molecules27030811_

Round 1

Reviewer 1 Report

Author had detail description of biology of SOD2. How ever in the context, there is no significant portion or reference to CD 44. Thus it might not surpport the title of this article

Author Response

Response to Reviewer 1 Comments

Point 1: Author had detail description of biology of SOD2. However, in the context, there is no significant portion or reference to CD44. Thus, it might not support the title of this article..

Response 1: We thank the reviewer for their valuable comment. It has to be emphasized that we have now included all our previous publications on the novel targets of CD44 that we have published from our microarray study conducted in 2006. The first target was validated with its novel signaling pathway and published in ONCOGENE (PMID: 16652145). Others followed later (PMID: 21718681, PMID: 17991717, PMID: 29121955, PMID: 23983821, PMID: 30443182).

Therefore, SOD2 follows the same scenario and ongoing experiments aim to validate and demonstrate the signaling pathways proposed in our Figure 2 (page 7, lines 254-255).

Last but not the least, we have rewritten the whole introduction to make these point clearer to the reader (pages 1-2, lines 32-50).

Reviewer 2 Report

This article addresses the potential role of SOD2 in cancer. The authors have provided a brief revision of SOD2 role in cancer and its potential involvement with CD44. This review is somewhat interesting considering the potential clinical implications of targeting SOD2 to tackle cancer progression and metastasis.

Although the authors state that will review the mechanisms and role of SOD2 in cancer, especially in breast cancer, it is not clear if they really what to review or just discuss their previous work and try convince us of their hypothesis (CD44 + SOD2). The authors should focus on the type of article chosen (literature Review) and not on their hypothesis. Otherwise, is better to provide a comment or a letter article. For instance on page 5, the authors describe what they found in their previous work and provided preliminary data (Fig3) to try validate the hypothesis that CD44 is a regulator of SOD2 expression. Moreover, the authors also state what they will do next. This is not appropriate for a Review. I will suggest remove all this section. The authors should improve the Review writing regarding SOD2 vs CD44.

Other points should be addressed:

  • Please, review all the manuscript regarding punctuation and formatting (e.g: line 10, 41, 92, 102, 104. 143, 192)
  • Figure 1 and 2 should be improved. The letter is very small and hard to read. The figures by itself are not attractive.
  • The authors described that SOD2 have a dual role in cancer (3.2). It will be nice if the authors could provide a sub-topic about the role of SOD2 as a tumor suppressor gene, since then only it is reviewed it role as an oncogene (3.3---)
  • Please, confirm is what is write at sentences line 115 and 124 (page 3) is what you what to describe.
  • The authors should explain better what is described at page 6 (line 190-193).
  • The section 3.4 is very poor, please improve. A table with potential inhibitors and/or clinical trials related with SOD2 will significantly improve the review.

Author Response

Response to Reviewer 2 Comments

Point 1: This article addresses the potential role of SOD2 in cancer. The authors have provided a brief revision of SOD2 role in cancer and its potential involvement with CD44. This review is somewhat interesting considering the potential clinical implications of targeting SOD2 to tackle cancer progression and metastasis.

Response: We thank the reviewer for their positive feedback. In addition, the new revised introduction shows clearly that we are indeed discussing the novel molecular link between CD44-HA interaction upstream and the activation of SOD2 expression (pages 1-2, lines 32-50).

Point 2: Although the authors state that will review the mechanisms and role of SOD2 in cancer, especially in breast cancer, it is not clear if they really what to review or just discuss their previous work and try convince us of their hypothesis (CD44 + SOD2). The authors should focus on the type of article chosen (literature Review) and not on their hypothesis. Otherwise, is better to provide a comment or a letter article. For instance, on page 5, the authors describe what they found in their previous work and provided preliminary data (Fig3) to try validate the hypothesis that CD44 is a regulator of SOD2 expression. Moreover, the authors also state what they will do next. This is not appropriate for a Review. I will suggest remove all this section. The authors should improve the Review writing regarding SOD2 vs CD44.

Response: We agree with the reviewer and figure 3 has been removed from the revised file (page 8). The section about our ongoing work has now been moved onto the section before the conclusion (pages 7-8, lines 256-273).

Other points should be addressed:

Point 3: Please, review all the manuscript regarding punctuation and formatting (e.g: line 10, 41, 92, 102, 104. 143, 192).

Response: We thank the reviewer for the suggestion and the grammatical errors have been corrected.

Point 4: Figure 1 and 2 should be improved. The letter is very small and hard to read. The figures by itself are not attractive.

Response: Since Figure 1 does not seem informative, we have removed the figure from the file (page 2, line 66). The Figure 2 has been revised (page 7, lines 254-255).

Point 5: The authors described that SOD2 have a dual role in cancer (3.2). It will be nice if the authors could provide a sub-topic about the role of SOD2 as a tumor suppressor gene, since then only it is reviewed it role as an oncogene (3.3---)

Response: We agree with reviewer and as indicated we have included a paragraph on the role of SOD2 as a TSG (page 4, lines 122-138). A table has also been incorporated within the text to show the mechanisms underlying the dual role of SOD2 in cancer (pages 4-5, lines 149-187).

Point 6: Please, confirm is what is write at sentences line 115 and 124 (page 3) is what you want to describe.

Response: We apologize for the typological error and the changes have been incorporated in the text (page 6, lines 196 and 203).

Point 7: The authors should explain better what is described at page 6 (line 190-193).

Response: As suggested by the reviewer, the sentence has been revised (page 8, lines 305-308).

Point 8: The section 3.4 is very poor, please improve. A table with potential inhibitors and/or clinical trials related with SOD2 will significantly improve the review.

Response: We have rewritten the section 3.4 (pages 8-9, lines 275-298; 341-354; 356-359) and as suggested, the table has been included (pages 9-10, lines 361-368).

Reviewer 3 Report

SOD2, a potential target underpinning CD44-promoted breast cancer progression

Alateyah et al. reviewed the effects of CD44-mediated SOD2 overexpression in breast cancer invasion and metastasis. Below are the comments regarding to this review.

Major Reviews

  • An extensive English grammar check is required.
  • Since this is a review paper, the main hypothesis of the paper, which was given in the Figure 3 (CD44-mediated SOD2 expression) can only be shown if it was published in a different research paper before. If this specific figure has not been published in a research article before, it should not be used in this paper.
  • Line 40 – incomplete sentence – This sentence should be reviewed again.
  • (Line 93-106) The dual role of SOD2 in tumorigenesis was shown by referencing different papers. However, it is not clear how SOD2 up/down-regulated in different cancer types. Cancer types/subtypes can be outlined with a table to make the role of SOD2 more clear in different cancer types.
  • (Line 124) – Incomplete sentence – ????
  • In Figure 3, you are showing that CD44 stimulation resulted in increased SOD2 expression and this effect can be rescued by siRNA knock down method. This result can be confirmed with an additional dual-glo based experiment. That type of promoter activity experiment will be more direct to make the same conclusion.

Minor Reviews

  • Period needs to be added at the end of the sentences after reference parenthesis.
  • Molecular structure of SOD2 protein can be shown with a figure under the title of ‘Structure of SOD2’.
  • Figure 1 should be redrawn properly. It does not look very informative. This schematic can be enriched with a more extensive figure.
  • (Line 113) – Acronyms should be written in long format at least once. It might night be clear what HA is for readers.
  • Incorrect punctuation marks were used in many places. They need to be corrected.

Author Response

Response to Reviewer 3 Comments

Point 1: An extensive English grammar check is required.

Response: We thank the reviewer for their comment and the English grammar check has been performed.

Point 2: Since this is a review paper, the main hypothesis of the paper, which was given in the Figure 3 (CD44-mediated SOD2 expression) can only be shown if it was published in a different research paper before. If this specific figure has not been published in a research article before, it should not be used in this paper.

Response: We agree with the reviewer and has suggested the figure 3 has been removed from the revised file (page 8).

Point 3: Line 40 – incomplete sentence – This sentence should be reviewed again.

Response: The section has been rewritten (pages 1-2, lines 32-50).

Point 4: (Line 93-106) The dual role of SOD2 in tumorigenesis was shown by referencing different papers. However, it is not clear how SOD2 up/down-regulated in different cancer types. Cancer types/subtypes can be outlined with a table to make the role of SOD2 clearer in different cancer types.

Response: We agree with reviewer and as indicated we have included a paragraph on the role of SOD2 as a TSG (page 4, lines 122-138). A table has also been incorporated within the text to show the mechanisms underlying the dual role of SOD2 in cancer (pages 4-5, lines 149-187).

Point 5: (Line 124) – Incomplete sentence – ????

Response: The sentence has been revised (page 6, line 204).

Point 6: In Figure 3, you are showing that CD44 stimulation resulted in increased SOD2 expression and this effect can be rescued by siRNA knock down method. This result can be confirmed with an additional dual-glo based experiment. That type of promoter activity experiment will be more direct to make the same conclusion.

Response: Based on the previous comment (comment# 2), figure 3 has been removed from the revised file (page 8).

Minor Reviews

Point 7: Period needs to be added at the end of the sentences after reference parenthesis.

Response: The reference style has been reformatted in the article. The period is now at the end of the sentences after reference parenthesis.

Point 8: Molecular structure of SOD2 protein can be shown with a figure under the title of ‘Structure of SOD2’.

Response: As suggested by the reviewer, the molecular structure of SOD2 protein has been included as Figure 1 (page 3, lines 90-91).

Point 9: Figure 1 should be redrawn properly. It does not look very informative. This schematic can be enriched with a more extensive figure.

Response: Since the figure does not seem informative, we have removed the figure from the file (page 2).

Point 10: (Line 113) – Acronyms should be written in long format at least once. It might night be clear what HA is for readers.

Response: As mentioned by the reviewer, full-forms for the acronyms have been provided for the first time within the text. A list of abbreviations is also now included in the end of the review (page 11, lines 379-395).

Point 11: Incorrect punctuation marks were used in many places. They need to be corrected.

Response: We thank the reviewer for the suggestion and the grammatical errors have been corrected.

Round 2

Reviewer 2 Report

The authors have revised the most important points raised by the reviewers and provided novel Tables that have substantially improved this revision work. I really appreciate that. Although interesting, this review is not a major contribution to the field. Overall the review is now acceptable however English grammar should be reviewed again by a native/professional English. 

Some minor points should be addressed:

  • line 53-60: this statement is out of the context. I suggest remove the text: "reactive species.....until....ROS detoxification".

  • Figure 1 does not bring any significance to the review.

  • line 134: The authors state that "few studies have shown SOD2 as a TSG". However, based on Table 1 the number of studies reporting SOD2 down-regulation in cancer is basically the same of those stating SOD2 as an oncogene. Please rephrase.

  • Topic 3.3 need major English revision. It is highly repetitive. Also the authors are always using "On the other hand" or "Moreover" or "also" or "In addition" all over the manuscript.

I have no other real comments.

Author Response

Response to Reviewer 2 Comments

Point 1: Overall, the review is now acceptable however English grammar should be reviewed again by a native/professional English.

Response: We thank the reviewer for their feedback. English grammar has been reviewed.

Minor Comments

 Point 2: line 53-60: this statement is out of the context. I suggest remove the text: "reactive species.....until....ROS detoxification".

Response: As suggested by the reviewer, the statement has been moved (page 2, lines 61-66).

Point 3: Figure 1 does not bring any significance to the review.

Response: The figure 1 has been removed from the review (page 2, line 88).

Point 4: line 134: The authors state that "few studies have shown SOD2 as a TSG". However, based on Table 1 the number of studies reporting SOD2 down-regulation in cancer is basically the same of those stating SOD2 as an oncogene. Please rephrase..

Response: The sentence has been rephrased (page 3, line 179).

Point 5: Topic 3.3 need major English revision. It is highly repetitive. Also, the authors are always using "On the other hand" or "Moreover" or "also" or "In addition" all over the manuscript.

Response: As indicated by the reviewer, the section has been revised for English (pages 4-5).

Reviewer 3 Report

accepted as it is after the review.

Author Response

Response to Reviewer 3 Comments

Point 1: Accepted as it is after the review.

Response: We thank the reviewer for their positive feedback.
